# Composite Nanostructure of Manganese Cluster and CHA-Type Silicoaluminaphosphates: Enhanced Catalytic Performance in Dimethylether to Light Olefins Conversion

**DOI:** 10.3390/nano11010024

**Published:** 2020-12-24

**Authors:** Guichen Ping, Kai Zheng, Qihua Fang, Gao Li

**Affiliations:** 1College of Science, Inner Mongolia Agricultural University, Hohhot 010018, China; pingaz@126.com; 2State Key Laboratory of Catalysis, Dalian Institute of Chemical Physics, Chinese Academy of Sciences, Dalian 116023, China; zhengkai@dicp.ac.cn (K.Z.); Cha_6299@163.com (Q.F.); 3University of Chinese Academy of Sciences, Beijing 100049, China

**Keywords:** manganese nanoclusters, SAPO-34, light olefins, heterogeneous catalysis, oxide-zeolite

## Abstract

Light olefins, especially ethylene and propylene, are important chemicals in petrochemical industries with an increasing demand and play an essential role in the global consumption. In this regard, there have been extensive studies to design efficient catalysts for the light olefins productions. In this study, we report a new protocol to induce Mn nanoclusters (MnNC) into the mesopore of a CHA-type silicoaluminaphosphates via a one-pot synthesis of MnNC@SAPO-34 catalysts. The catalysts are characterized by a series of technology, such as TEM, XRD, NH_3_-TPD, ^27^Al MAS NMR, ICP-MS, XPS, and as well as N_2_-physical adsorption methods. The Mn nanoclusters of Mn_2_O_3_ and MnO_2_ species are well dispersed in the framework of the SAPO-34 silicoaluminaphosphates, modifying the porosity and acidic property of the SAPO-34: Giving rise to more mesoporous and improving the acid density. The MnNC@SAPO-34 catalysts exhibit decent 100% conversion and 92.2% olefins selectivity in the dimethyl ether to olefins (DTO) reactions, which is considerably higher than that for SAPO-34 silicoaluminaphosphates (79.6% olefins selectivity). The higher olefins selectivity over the MnNC@SAPO-34 is deemed to associate with the strong acid density and intensity of the silicoaluminaphosphates. Further, the Mn particles largely improve silicoaluminaphosphates’s durability.

## 1. Introduction

Light olefins, especially ethylene and propylene, are important chemicals in petrochemical industries with an increasing demand and play an essential role in the global consumption [1]. Conventionally, the light olefins are often produced from steam cracking and fluid catalytic cracking of naphtha. With diminishing oil reserves, it is desirable to develop technologies from alternative non-petroleum resources (e.g., coal, biomass, natural gas) via syngas chemistry. Methanol/dimethylether (DME) to olefins (M/DTO) is an important process for non-oil route to synthesize light olefins from syngas or coal in C1 chemistry [2,3,4]. The M/DTO reactions often are catalyzed over the silicoaluminophosphate (e.g., SAPO-34), which has been proven to be the most attractive candidate for industrial production in China in 2010 [5,6]. The carbon depositions are quickly formed during M/DTO process, which will block the active sites and the small channels of the silicoaluminaphosphates. Then the activity of the SAPO-34 catalysts will rapidly decrease and deactivate in a few hours (e.g., <1 h) [7,8]. Therefore, tailoring the olefins distribution, especially to ethylene and propylene, and the durability of the SAPO-34 catalyst remain a great challenge and issue for methanol or DME to light olefins process.

In the previous literatures, the incorporation of heteroatoms into the zeolite’s framework can promote the product distribution in the methanol to hydrocarbon process, which could be attributed to the modifications of the acidic strength, the acid distribution, the pore size, and the channel length of zeolite by the incorporated heteroatoms [9,10,11,12,13]. Liu and Su et al. directly incorporated a serial of metal ions into the SAPO-34 molecular sieve using the metal salts, which slightly improved the selectivity for ethylene and propylene [9]. They predicted that the slightly improved selectivity was caused by the formed stronger acid sites though the Mn incorporation. The catalytic activity and selectivity also can be promoted by the introduction of Mn in the Fischer–Tropsch Synthesis [14]. Next, the MnO_2_ oxides were mixed with SAPO-34 via simply mechanical agitation, which also was found to improve the selectivity for light olefins [15]. The manganese oxides aggregated to a size of 50–75 nm. However, the incorporation of Mn nanoclusters (the size <3 nm) is rarely investigated in the previous literatures.

Very recently the small-sized metal oxide particles are found to largely improve the catalytic performance of the decorated nanocomposites during the chemical transformations [16,17,18].

In our recent work, the mesoporous HZSM-5 zeolite incorporated Ni nanoclusters of 1.2–2.7 nm showed better catalytic activity and selectivity in the DME to gasoline (DTG) process [19]. Herein, we demonstrate a new protocol to induce Mn nanoclusters encapsulated in the mesoframework of SAPO-34 molecular sieve via a one-pot reaction of (Mn^II^L_2_) nanoclusters within the presence of an aqueous solution of aluminum isopropoxide, phosphoric acid, diethylamine, and TEOS. The Mn nanoclusters converted to ultra-small (MnO_2_)_n_ and (Mn_2_O_3_)_n_ species during 550 °C annealing. To our best knowledge, it is the first time to report such small MnO_2_/Mn_2_O_3_ nanoclusters into the silicoaluminaphosphates. The porosity and acidic property of the silicoaluminaphosphates are largely modified by the incorporation of Mn nanoclusters. For the DTO process in the presence of hydrogen gas, the MnNC@SAPO-34 gives a 100% DME conversion with a high selectivity for C2-4 = olefins (92.2%), which is much better than that catalyzed over synthesized SAPO-34 (79.6% C2-4 = selectivity). Simultaneously, the MnNC@SAPO-34 catalysts exhibit 10 times longer durability (over 5 h of operation) than the SAPO-34 (ca. 30 min), which is due to the inhibition of carbon deposition production.

## 2. Experimental

### 2.1. Chemicals

All commercial chemicals are purchased in analytical grade and directly used without any further purifications. Manganese acetate (99%), tetraethyl orthosilicate (TEOS, 99%), aluminum isopropoxide (98%), sodium borohydride (99.9% metals basis), phosphoric acid (98%), diethylamine (99%), (3-mercaptopropy)trimethoxysiliane (MPTS, 98%) are purchased from Adamas China. Nanopure water with 18.2 MΩ cm^−1^ resistance was acquired from a NANOpure Di-water TM system (Barnstead Company, Boston, MA, USA). All glassware was cleaned with an aqua regia solution of HCl:HNO_3_ 3:1 (*v*/*v*), and thoroughly rinsed with copious Nano-pure water. It was dried in an oven at 60 °C overnight prior to use.

### 2.2. Preparation of MnNC@SAPO-34 Catalysts

The MnNC@SAPO-34 catalyst was synthesized according to our previous method [8,9] It contains two main steps. Typically, in the first step 0.16 g manganese acetate was dissolved in a mixed solution (5 mL water and 15 mL ethanol). And then, 150 µL (3-mercaptopropy)trimethoxysiliane (H-SC_3_H_6_Si(OCH_3_)_3_, MPTS) was added in the mixed solution, and the reaction mixture was stirred for 10 min to give manganese nanoclusters (Mn_n_(MPTS)_m_), which was named as solution A for the next step. In the second step, 2.939 g H_3_PO_4_ and 6.127 g aluminum isopropoxide was dissolved in 22 mL water. After stirring 20 min, 4.388 g diethylamine was added in the aqueous solution. Ten minutes later, 1.25 g tetraethyl orthosilicate (TEOS) was added and stirred for another 10 min. And then, the solution A was fast added into the system. After 10 min the reaction solution was sealed in a reactor and heated at 200 °C for 36 h. The reactor was cooled to room temperature. The solid sample was filtrated and washed with water for three times. It was dried at 80 °C overnight. Finally, the MnNC@SAPO-34 catalyst was obtained via calcination at 550 °C for 4 h in air. For comparison, the plain SAPO-34 silicoaluminaphosphates was obtained using same process without the addition of manganese nanoclusters. Of note, the particle size of the foreign transition metal species is >50 nm using the post-treatment of SAPO-34 with transition metal salts [19].

### 2.3. Characterization

Powder X-ray diffraction spectra were acquired on an X’Pert PRO diffractometer (PANalytical) with CuKα radiation of 0.15406 nm at 40 kV and 40 mA to identify the phase structure of the samples. The XRD data was recorded in a 2θ range of 5 to 90° with a step size of 0.0334° and a counting time of 47 s per step. The ^27^Al magic angle spinning nuclear magnetic resonance (MAS NMR) was performed on the Bruker 400M (AVANCE III 400 MHz). X–ray photoelectron spectra (XPS) were recorded with an ESCALAB 250 Xi spectrometer (Thermofisher, Waltham MA, USA) using an Al Kα radiation source operated at an accelerating voltage of 15 kV. The charge effect was corrected by adjusting the binding energy of C1s to 284.6 eV. Inductively coupled plasma-mass spectrometry was recorded on PerkinElmer ICP-MS NexION 300D. The MnNC@SAPO-34 samples were decomposed by aqua regia solution. Temperature-programmed oxidation mass spectrometry (TPO-MS) was recorded on the quadrupole mass spectrometer OmniStarTM GSD 301 (Pfeiffer Vacuum, Hessen, Germany) to detect and analyze signal of H_2_O (*m*/*z*: 18) and CO_2_ (*m*/*z*: 44). 40 mg sample was pretreated at 200 °C for 2 h in He flow to eliminate the products, and then it was heated from 30 °C to 900 °C with a heating rate 10 °C min^−1^ under a flow of 5 *v*% O_2_/95 *v*% Ar. Transmission electron microscope (TEM) was performed on JEM-2100 instrument with an acceleration voltage of 200 kV. The N_2_-physical adsorption of 100 mg samples was analyzed by Micromeritics ASAP 2020 surface area analyzer (Quantachrome, Boynton Beach, FL, USA). The acidity of the samples was examined by NH_3_ temperature-programmed desorption (NH3-TPD) in an auto-catalytic adsorption system with a quartz tubular reactor (AMI-200, Zeton Altamira) and an on-line mass analyzer (OmniStarTM, GSD301, Hessen, Germany). The samples were pretreated at 500 °C in argon flow for 1 h, and then they were cooled to 50 °C. Ammonia (20 *v*% NH3/Ar) was introduced at a flow rate of 30 mL/min for 0.5 h at 50 °C, and blown by argon. The reactor temperature was programmed to increase at a ramp rate of 10 °C/min, and the amount of ammonia (*m*/*z* = 17) in effluent was measured and recorded as function of temperature. Thermogravimetric analysis (TGA) was performed on a TGA 2 STARe system (Mettler Toledo, Greifensee, Switzerland). The spent catalysts of ~10 mg were used in the test, which was conducted in air with a 50 mL/min gas flow and a 10 °C/min heating rate.

### 2.4. Catalytic Evaluation in the DTO Reaction

A pressurized flow-type reaction apparatus with a fixed bed reactor was used for this DTO reaction. The fixed bed reaction system is comprised of an electronic temperature controller and a tubular reactor (inner diameter: 12 mm). Before the reaction, the silicoaluminaphosphate catalysts of 150 mg were activated at 300 °C for 4 h in a nitrogen flow. Then the reactant gas of DME/H_2_/N_2_ (14.4/30.6/55.0, *v*/*v*), was introduced into the reaction system. The DTO reactions were carried out under the catalytic conditions of 350 °C, 1.5 MPa, and a reaction gas flow rate of 3000 mL h^−1^ g^−1^. All the products from the tubular reactor were introduced to and analyzed by the online GC apparatus in the gaseous state. The CO, CO_2_, CH_4_, and N_2_ gas were analyzed using a GC system (SHIMADZU, GC-8A) equipped with a thermal conductivity detector (TCD), and the other hydrocarbon compounds were analyzed using another GC system (VARIAN 3800) equipped with a flame ionization detector (FID).

## 3. Results and Discussion

### 3.1. Synthesis and Characterization of the MnNC@SAPO-34 Catalysts

The SAPO-34-capping manganese cluster catalyst (noted as MnNC@SAPO-34, hereafter) was synthesized via an one-pot reaction of Mn_n_(MPTS)_m_ nanoclusters in the presence of a mixed aqueous solution of aluminum isopropoxide, phosphoric acid, diethylamine, and TEOS. And then the as-prepared samples were annealed at 550 °C in air for 4 h to eliminate the organic thiolates and surfactants. ICP-MS analysis suggested that the MnO_x_ content of MnNC@SAPO-34 is ca. 0.51 wt%. For comparison, we also prepared the free SAPO-34 silicoaluminaphosphates using the same synthetic method. These as-obtained SAPO-34 and MnNC@SAPO-34 catalysts were characterized by XRD, N_2_ adsorption/desorption isotherms, as well as NH_3_-TPD.

The powder X-ray diffraction (XRD) analysis of the SAPO-34 and MnNC@SAPO-34 presented in Figure 1a. The XRD patterns of two samples matched well with typical patterns (2θ = 9.5, 12.9, 16.1, 17.9, 20.7, 25.0, 26.0, 30.7 and 31.2°) of a CHA structure (PDF#01-087-1527), indicating that these samples possess the skeletal structure of a SAPO-34 silicoaluminaphosphates with a three-dimensional channel structure. No diffraction lines of MnO_x_ particles was found, which is primarily because of the well dispersion of Mn particles in small sizes or the lower MnOx loading (ca. 0.51 wt%). The porosity of SAPO-34 and MnNC@SAPO-34 was characterized by measuring N_2_ sorption isotherms at 77 K. The nitrogen adsorption isotherms of samples showed the presence of a hysteresis loop, suggesting the formation of a mesoporous structure (secondary pores). The N_2_ adsorption–desorption isotherms and the pore distribution of the MnNC@SAPO-34 are different with SAPO-34 zeolite (Figure 1b, black and blue curves), suggesting that the physicochemical properties of the MnNC@SAPO-34 and SAPO-34 are distinct. The SAPO-34 silicoaluminaphosphates exhibit larger BET than the MnNC@SAPO-34 (188.40 vs. 150.26 cm^2^/g, Table 1). Although the two samples have the same total volume (Vtotal, 0.12 cm^3^/g), the MnNC@SAPO-34 possesses isometric microporous and mesoporous volume (Vmic = Vmes = 0.06 cm^3^/g), and the SAPO-34 shows more microporous volume (0.08 cm^3^/g), Table 1. And the MnNC@SAPO-34 exhibits larger mesopore than the corresponding SAPO-34 (12.1 nm vs. 6.1 nm), which is mainly caused by the introduction of the MPTS protected Mn nanoclusters during the one-pot synthesis.

The acidic property of SAPO-34 and MnNC@SAPO-34 was determined by NH_3_-TPD tests. The intensive peaks at~130 °C can be assigned to the NH_3_ desorption from the weak acidic sites of the silicoaluminaphosphates, and the~400 °C peaks were attributed to the NH_3_ desorption over the strong acidic sites. As depicted in Table 1, the acid density, both the strong and weak acidic sites, of MnNC@SAPO-34 were much higher than the corresponding pure SAPO-34. Therefore, the in-situ introduction of the Mn nanoclusters largely tuned the physical property (porosity and acidic sites) of the silicoaluminaphosphates. The MnNC@SAPO-34 composite is proposed to improve the catalytic performance in the DTO reaction.

Further, the morphology and the distribution of the small sized manganese clusters in the MnNC@SAPO-34 samples were investigated by STEM energy-dispersive X-ray spectroscopy (EDS, Figure 2 and Figure 3). The manganese, aluminum, silicon, and oxygen elements are well overlapped in the STEM EDX analysis, strongly indicating that the manganese clusters are uniformly located into the framework of the SAPO-34. Furthermore, the sulfur and carbon elements are absent, manifesting that the MPTS ligands and surfactant are all removed during the 550 °C annealing process.

To confirm the chemical state of the surface composition in the MnNC@SAPO-34 samples, XPS was conducted with results compiled in Figure 4. As seen in the Figure 4a–c, the XPS peaks at 75.1, 102.5, and 134.5 eV are contributed to characteristic Al_2_O_3_, SiO_2_, and P_2_O_5_ species [20]. Meanwhile, the Mn 2p doublet are found at 654.0 and 641.8 eV with an energy difference of~12.2 eV (Figure 4d), which corresponds to Mn2p_1/2_ and Mn2p_3/2_ spin-orbit-split doublet peak of manganese oxides [20,21]. The deconvoluted Mn 2p spectrum can be assigned to main two contributions; the peaks at 641.9 and 653.9 eV could be assigned to Mn^3+^, and the peaks at 643.9 and 656.3 eV belong to Mn^4+^ species. These results are in agreement with the literatures [20,21,22]. The quantitative analysis of relative content of the manganese ions (Mn^4+^ and Mn^3+^) on the surface is obtained by calculating the area ratios of corresponding species derived from the raw XPS data. The ratio of the Mn^3+^ to Mn^4+^ is 3:1, meaning that most of the Mn^II^L_2_ precursors are oxidized to Mn_2_^III^O_3_ and Mn^IV^O_2_ species during the annealing process at 550 °C in air. The Mn_2_O_3_ and MnO_2_ were reported as the catalytic site in the syngas conversion to light olefins.

Figure 5 gives the ^27^Al MAS NMR spectra of the synthesized SAPO-34 and MnNC@SAPO-34. The two peak signals at 36.9 and −9.8 ppm are found in the ^27^Al MAS NMR analysis of SAPO-34 (Figure 5, black line), which can be assigned to tetrahedrally coordinated framework Al species (Al^tetra^) and octahedrally coordinated extraframework ones (Al^oct^) respectively [23]. The incorporation of Mn nanoclusters make difference on the chemical shifts. The corresponding ^27^Al NMR peaks of the MnNC@SAPO-34 are red-shifted to 44.2 and −9.4 ppm, respectively. And the molar ratio of Al^tetra^ to Al^oct^ is 2.33 for free SAPO-34 and 1.14 for the MnNC@SAPO-34 catalysts, respectively, which is a strong indication that more extraframework Al species formed in the MnNC@SAPO-34 catalysts. Of note, the broaden ^27^Al NMR peaks of MnNC@SAPO-34 samples are another indication of the existent of magnetic particle of Mn_2_O_3_ and MnO_2_ species, and the related amounts of Al^tetra^ for the MnNC@SAPO-34 and SAPO-34 is not clear, as the sampling frequency is different.

### 3.2. Catalytic Performance of the Catalysts

The catalytic results over the SAPO-34 silicoaluminaphosphates and MnNC@SAPO-34 are detailed in Figure 6. The SAPO-34 and MnNC@SAPO-34 catalysts exhibited distinct catalytic performance under the identical reaction conditions (a GHSV of 3000 mL g^−1^ h^−1^ at the reaction temperature of 350 °C). Both the SAPO-34 silicoaluminaphosphates and MnNC@SAPO-34 catalysts gave 100% DME conversion at the beginning (ca. 32 and 40 min, Figure 6a). But the catalytic activity of SAPO-34, based on the conversion of DME, was quickly decreased to 99% at 68 min and to 83% at 104 min (Figure 6a, black line), consistent with the previous reported literatures [4]. In contrast, the MnNC@SAPO-34 catalyst maintained the 100% DME conversion over 192 min, and then it was gradually deactivation (93% conversion at reaction time of 330 min), Figure 6a, blue line. The deactivation is mainly due to that the strong acid sites of catalysts are covered by the formed carbon (vide infra). Hence, the MnNC@SAPO-34 catalysts showed better durability and longer lifetime than the corresponding SAPO-34 silicoaluminaphosphates.

With respect to the product distribution in the light olefins (e.g., C_2_H_4_, C_3_H_6_, and C_4_H_8_), MnNC@SAPO-34 and SAPO-34 catalyst showed a similar selectivity at the induction period. It is worth noting that the carbon balance is close to 100%, and the by-products are CO and light alkanes, meaning that the DME is decomposed over the surface of the MnNC@SAPO-34 during the initial 2 h reactions. A 54.6% olefins selectivity (containing 18.5% C_2_H_4_, 22.1% C_3_H_6_, and 14.0% C_4_H_8_) over SAPO-34 and 50.0% selectivity for olefins comprising of 16.0% C_2_H_4_, 20.0% C_3_H_6_, and 14.0% C_4_H_8_ over the MnNC@SAPO-34 catalysts were found at 32 and 40 min on stream (Figure 6b,c). After the short induction period, the selectivity of olefins over the SAPO-34 silicoaluminaphosphates and MnNC@SAPO-34 was gradually improved to ca. 79.6% (containing 32.2% C_2_H_4_, 34.8% C_3_H_6_, and 12.6% C_4_H_8_ at 104 min) and 92.2% (36.5% C_2_H_4_, 45.2% C_3_H_6_, and 10.5% C_4_H_8_ at 192 min). Of note, olefins products are the most valuable chemicals during the DTO process. It is interesting that the selectivity for butylene over the SAPO-34 and MnNC@SAPO-34 maintained in the range of 12.6% to 14.0% and 10.5% to 16.5% during the DTO reactions, Figure 6b,c. In all, the MnNC@SAPO-34 catalysts exhibited much better catalytic performance (including the activity and product distribution) than the corresponding SAPO-34 silicoaluminaphosphates, which may be due to the synergism of the SAPO-34 and Mn clusters during the DTO reactions.

Finally, TPO-MS and TGA was performed to study the deactivation of spent MnNC@SAPO-34 and SAPO-34 catalysts in the DTO reactions. As seen in Figure 7a, TGA showed that the carbon deposition reaches to ca. 4.4 wt% and 2.6 wt% of the spent SAPO-34 (after 2 h) and MnNC@SAPO-34 (after 8 h) catalysts, respectively. Of note, the DME conversion over the spent SAPO-34 and MnNC@SAPO-34 decreased to 60–70%. Further, the obvious H_2_O and CO_2_ signals were detected around 450–750 °C by mass spectrometer in the used SAPO-34 and the used MnNC@SAPO-34 catalysts, which can be assigned to the combustion of carbon deposition (C_x_H_y_ hydrocarbons). The intensity of H_2_O and CO_2_ signals arising from the used SAPO-34 is much stronger than these of MnNC@SAPO-34 catalyst, implying that the MnNC@SAPO-34 catalyst can inhibit the formation of carbon deposition during the DTO reaction. It is mainly due to the large mesoporous for mass transfer and the high concentration of the extraframework Al species and acid density in the MnNC@SAPO-34 [5,24,25,26,27].

## 4. Conclusions

In summary, a new synthetic protocol is applied for the incorporation of manganese nanoclusters into a CHA-type silicoaluminaphosphates via a one-pot synthesis process. The Mn nanoclusters are uniformly dispersed in the framework of MnNC@SAPO-34 catalysts. The MnNC@SAPO-34 exhibits more mesoporous and higher acid density and generates more octacoordinated Al species. During the DTO reactions, MnNC@SAPO-34 catalysts show considerably higher catalytic performance, including olefins selectivity and durability, which is due to the inhibition of carbon deposition production. In all, this study will give some cue to the development and design of efficient metal cluster-modified porous zeolites for the other C1 conversions.

## Figures and Tables

**Figure 1 nanomaterials-11-00024-f001:**
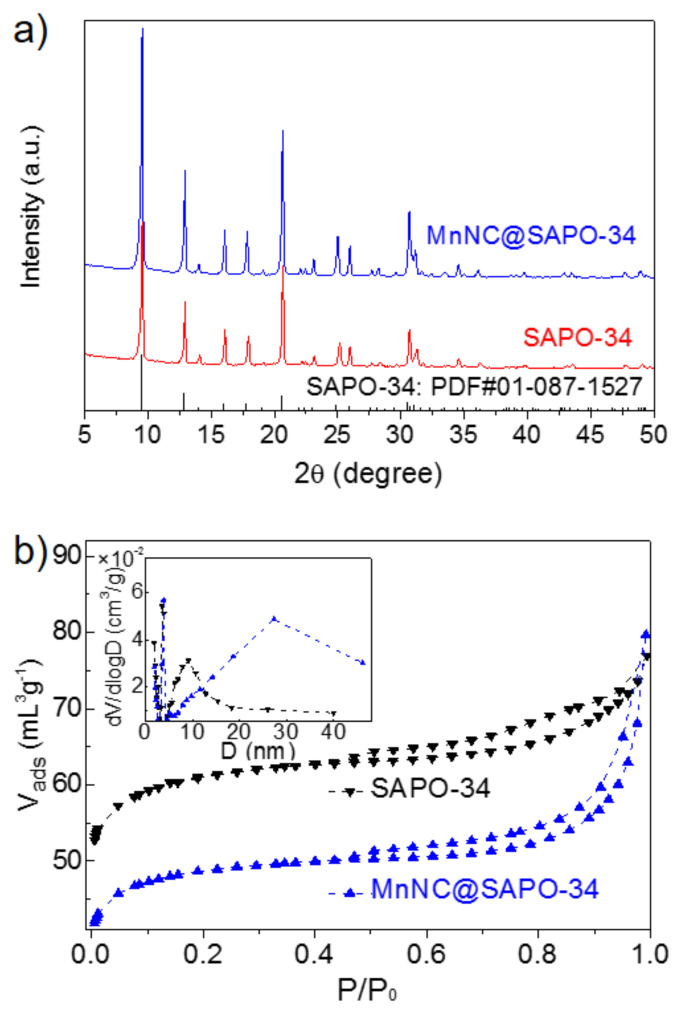
Powder XRD patterns (**a**) and N_2_ adsorption/desorption isotherms (**b**) of SAPO-34 silicoaluminaphosphates and MnNC@SAPO-34 samples. Inset in (**b**): Pore-size distribution.

**Figure 2 nanomaterials-11-00024-f002:**
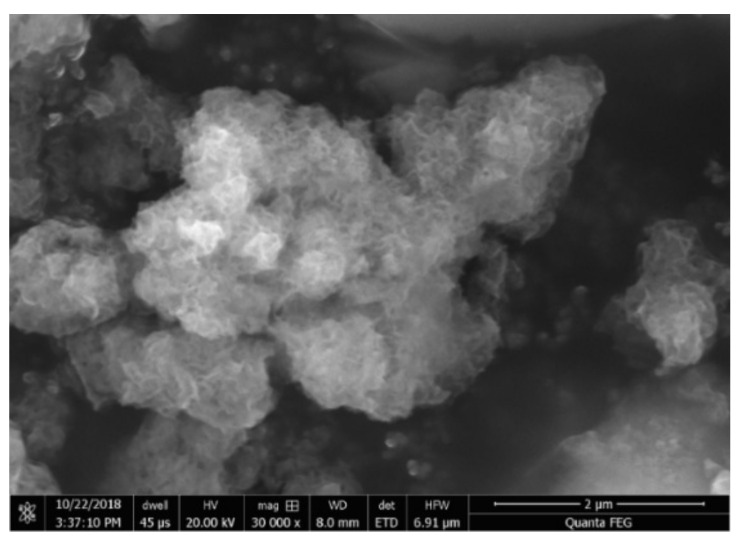
SEM image of MnNC@SAPO-34.

**Figure 3 nanomaterials-11-00024-f003:**
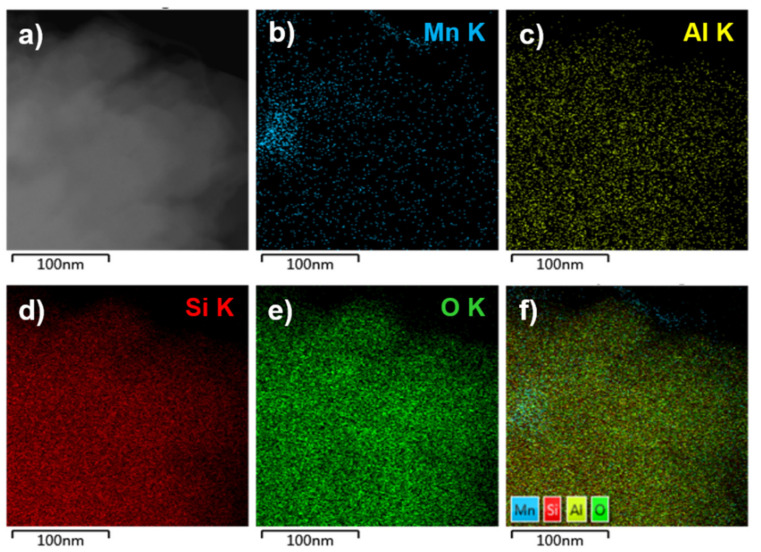
TEM image (**a**) and elemental mapping of Mn (**b**), Al (**c**), Si (**d**), O (**e**), and all elements (**f**) of the MnNC@SAPO-34 sample.

**Figure 4 nanomaterials-11-00024-f004:**
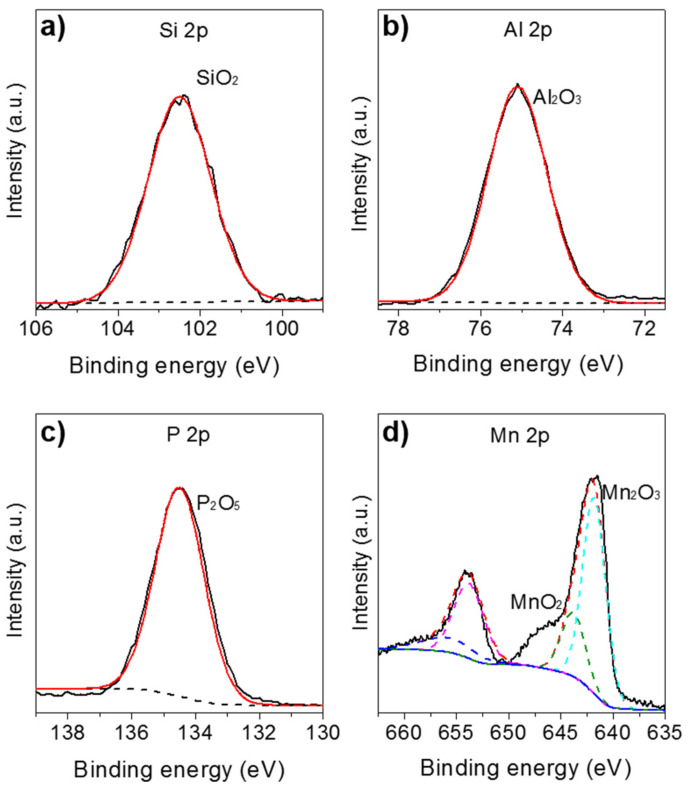
Si 2p (**a**), Al 2p (**b**), P 2p (**c**), and Mn 2p (**d**) XPS spectra of the MnNC@SAPO-34 catalysts.

**Figure 5 nanomaterials-11-00024-f005:**
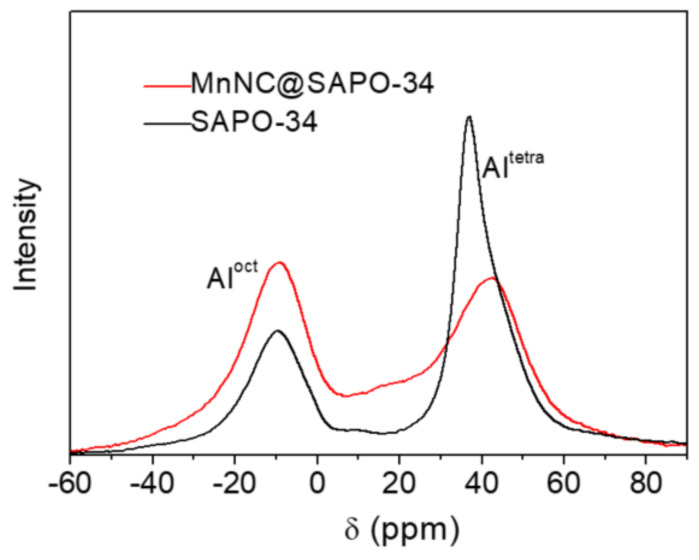
Solid-state ^27^Al NMR spectra of SAPO-34 and MnNC@SAPO-34 catalyst. The ^27^Al NMR peaks at 36.9 and 44.2 ppm are attributed to tetrahedrally coordinated Al (framework Al, noted as Al^tetra^), and other peaks at -9.8 and -9.4 ppm are assigned to octahedrally coordinated Al (extraframework Al, Al^oct^).

**Figure 6 nanomaterials-11-00024-f006:**
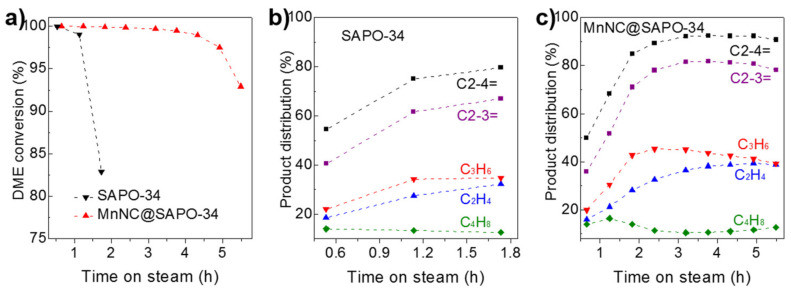
(**a**) Catalytic activity, based on dimethylether (DME) conversion, over the SAPO-34 silicoaluminaphosphates and MnNC@SAPO-34 catalysts in the dimethyl ether to olefins (DTO) process. Product distribution (selectivity for light olefins of C2-4=, C2-3=, C_2_H_4_, C_3_H_6_, and C_4_H_8_) catalyzed on the (**b**) SAPO-34 and (**c**) MnNC@SAPO-34. Reaction conditions: 1.5 MPa (14.4 *v*% DME, 30.6 *v*% H_2_, 55.0 *v*% N_2_), GHSV = 3000 mL g^−1^ h^−1^, 150 mg catalyst, 350 °C.

**Figure 7 nanomaterials-11-00024-f007:**
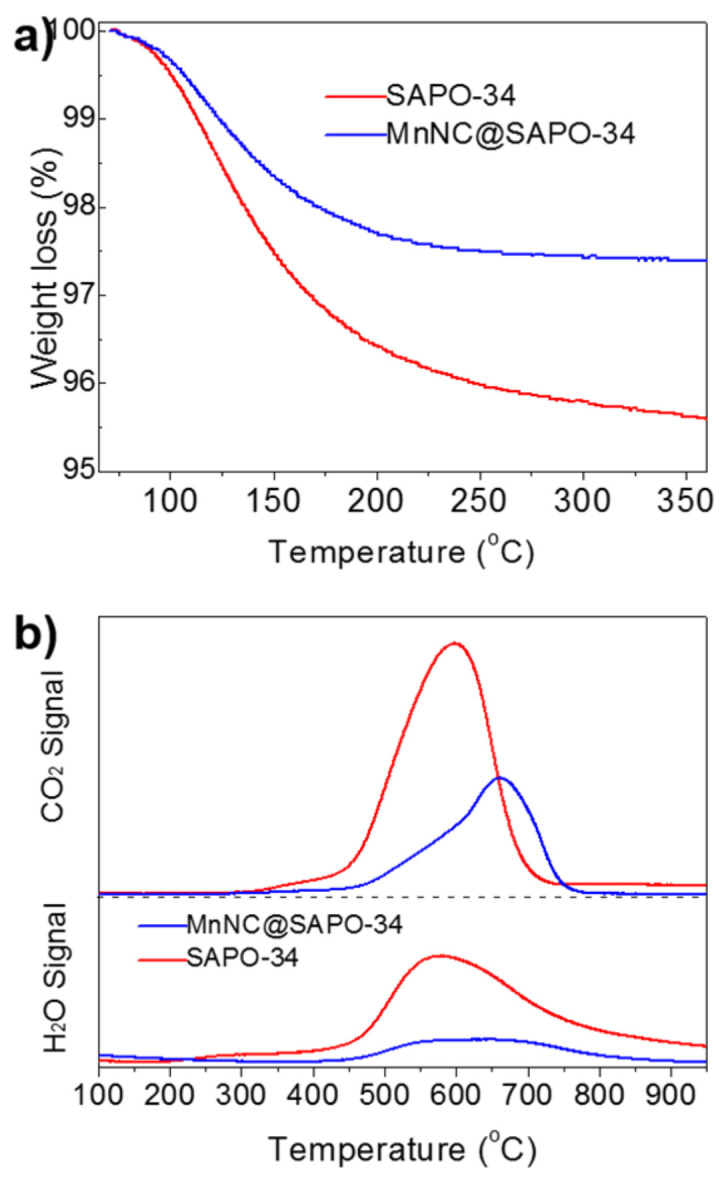
TGA (**a**) and TPO-MS (**b**) of carbon deposition of the spent SAPO-34 (after 2 h) and MnNC@SAPO-34 (after 8 h) catalysts when the DME conversion decreased to 60–70%.

**Table 1 nanomaterials-11-00024-t001:** Physical property and acid density of SAPO-34 silicoaluminaphosphates and MnNC@SAPO-34 composites.

Sample	Physical Structures					Acid Density (mmol/g)	
	SBET (cm^2^/g) ^a^	Vtotal (cm^3^/g) ^b^	Vmic (cm^3^/g) ^c^	Vmes (cm^3^/g) ^d^	Dpore (nm) ^e^	Weak Acid Site	Strong Acid Site
SAPO-34	188.40	0.12	0.08	0.04	6.1	0.415	0.702
MnNC@SAPO-34	150.26	0.12	0.06	0.06	12.1	0.568	0.953

^a^ BET method. ^b^ Vtotal was measured at P/P_0_ = 0.98. ^c^ Using t-plot method. ^d^ V_mes_ = V_total_ − V_mic_. ^e^ BJH desorption average pore diameter.

## Data Availability

The data that support the findings of this study are available from the corresponding author, [Gao Li], upon reasonable request.

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
