# Peer review of "Composite Nanostructure of Manganese Cluster and CHA-Type Silicoaluminaphosphates: Enhanced Catalytic Performance in Dimethylether to Light Olefins Conversion"

_nanomaterials, 2020, doi:10.3390/nano11010024_

Round 1

Reviewer 1 Report

This article concerns with the dimethyl ether to olefins reaction on the Mn nanoclusters induced SAPO-34 zeolite. The olefin selectivity and the stability was excellent. The results are new and original. The research field is suitable to the “nanomaterials”. However, the discussions of the reaction are weak. It is not clear that the effect of the one-pod synthesis of MnNC@SAPO-34. I think this article requires minor revision for publication to the “nanomaterials”.

In order to show the effects of the one-pod synthesis of MnNC@SAPO-34, please discuss about the SAPO-34 with the post-treatment of Mn.

1P line 26: Authors claimed that the higher olefin selectivity is due the strong acid of the MnNC@SAPO-34. However, the weak acid of the MnNC@SAPO-34 is larger than that of the SAPO-34 shown in Table 1. There is a possibility of the strong acid is covered by the carbon. Thus, NH3-TPD results for the spent MnNC@SAPO-34 should be required for the effects of the acid.

7P line 206: The amounts of Altetra for the MnNC@SAPO-34 are smaller those that for the SAPO-34. However, the amounts of acid for the MnNC@SAPO-34 shown in Table 1 are larger than those for the SAPO-34. Please explain this contradiction.

8P line 246: The steady state of the DTO reaction for the MnNC@SAPO-34 was very slow (Fig. 6c). The surface state of the MnNC@SAPO-34 must be changing during the initial 2 h of the DTO reaction. Please show the reason(s) of the initial situation of the reaction. Especially for the initial 2h of the reaction, please show the carbon balance and the by products for the discussions.

Reviewer 2 Report

After reading the manuscript, I regard it as a good work worthy of publication in this journal. I only have some remarks to propose.

Firstly, on lines 171 and 173, the temperatures corresponding to the weak and strong acid sites must be wrong and they should be corrected.

Secondly, Figure 3.b) is very dark and it is hard to tell the Mn moieties. I think it should be certainly cleared.

Finally, I have missed the measurement of the amounts of Bronsted and Lewis acid sites. I agree with the authors that the enhanced mesoporosity of the MnCHA improves its performance but I also think it contains fewer strong Bronsted acid sites and more Lewis acid sites and this may relevant to this reaction.

Anyway, the authors should provide at least some literature about the nature of the acid sites responsible for this reaction.
